# Antimicrobial Effects of *_L_*-Chg_10_-Teixobactin against *Enterococcus faecalis* In Vitro

**DOI:** 10.3390/microorganisms10061099

**Published:** 2022-05-26

**Authors:** Alaa Jarkhi, Angeline Hui Cheng Lee, Zhenquan Sun, Mingxin Hu, Prasanna Neelakantan, Xuechen Li, Chengfei Zhang

**Affiliations:** Faculty of Dentistry, University of Hong Kong, Hong Kong 999077, China; 2018alaajarkhi@gmail.com (A.J.); bollies4@hku.hk (A.H.C.L.); sunzhq3@connect.hku.hk (Z.S.); humingxin2009@hotmail.com (M.H.); prasanna@hku.hk (P.N.); xuechenl@hku.hk (X.L.)

**Keywords:** antibiotic, biofilm, *Enterococcus faecalis*, teixobactin analogue

## Abstract

**Objective:** Teixobactin and its analogues are a new class of antibiotics that have no detectable bacterial resistance. This study was designed to determine the antibacterial and antibiofilm activities of a novel teixobactin analogue, *_L_*-Chg_10_-teixobactin, against two strains of *Enterococcus faecalis (E. faecalis)*. **Materials and Methods:** The efficacy of *_L_*-Chg_10_-teixobactin against two strains of *E. faecalis* (ATCC 29212 and 47077) was determined using Clinical and Laboratory Standards Institute methods. *_L_*-Chg_10_-teixobactin was prepared at a stock concentration of 1 mg/mL in 5% DMSO. The minimum inhibitory concentration (MIC) was calculated using a two-fold serial broth dilution method, utilizing a 96-well plate. The minimum bactericidal concentration (MBC) was determined by plating the bacteria onto agar to define the concentration that resulted in 99.9% of bacterial death. Ampicillin was used as the control. The effect of *_L_*-Chg_10_-teixobactin on the inhibition of ATCC 47077 strain biofilm formation was determined by measuring the minimum biofilm inhibitory concentration (MBIC) using the safranin assay, while the eradication of the preformed biofilm was determined by measuring the minimum biofilm eradication concentration (MBEC) using the XTT assay. For nonlinear data, the log dose–response curve was plotted to calculate the optimum concentration using Excel (version 16.51, Microsoft^®^ excel. 2021, Microsoft Corporation, Reymond, WA, USA). The data are presented as mean ± standard deviation (SD). **Results:** The MIC and MBC values of *_L_*-Chg_10_-teixobactin against both strains of *E. faecalis* were 0.8 μg/mL. The MIC of ampicillin was 1.25 μg/mL for ATCC 29212 and ranged from 1.25 to 5 μg/mL for ATCC 47077. The MBC of ampicillin for ATCC 29212 and ATCC 47077 was 10 and 20 μg/mL, respectively. The MIC and MBC of ampicillin were much higher compared with those of *_L_*-Chg_10_-teixobactin. The MBEC_80_ of *_L_*-Chg_10_-teixobactin was 4.60 μg/mL for ATCC 47077, which was much lower than that of ampicillin (20 μg/mL). **Conclusions:**
*_L_*-Chg_10_-teixobactin demonstrated potent antibacterial and antibiofilm effects against *E. faecalis*, suggesting its potential role an effective antibacterial and antibiofilm agent in endodontic treatment.

## 1. Introduction

*Enterococcus faecalis* (*E. faecalis*) is a Gram-positive, facultative anaerobic, and biofilm-forming pathogen that is associated with endodontic and systemic infections [1,2]. It can both survive and form a biofilm in harsh environments, such as obturated root canals [3,4]. This is evidenced by the failed cases in which root canals have been adequately obturated [3,4]. *E. faecalis* is much more frequently isolated in cases with failed endodontic therapy as a single organism or a major component of mixed flora, while it is only occasionally detected in primary endodontic infections, suggesting its major role in causing persistent infections after root canal treatment [5]. In these failed endodontic cases, the bacterial biofilms can be found in the main canals, as anatomical irregularities such as isthmus and apical ramifications, and in the extraradicular surface [6,7]. Although the current chemo-mechanical strategies supplemented with various activation approaches, such as sonic and ultrasonic activation, are much more effective in removing biofilms from the main canals, they are largely ineffective at removing biofilms in hard-to-reach areas such as anatomical irregularities and the extraradicular surface [8,9]. Therefore, novel strategies are urgently needed to tackle this issue.

Intracanal medicament has been advocated as an essential means of eliminating residual bacteria after canal shaping and cleaning, while modern endodontics does not support the necessity of its routine use as the clinical outcomes of single- and multiple-visit root canal therapy are similar [10]. Nevertheless, the frequency and volume of residual bacteria found in root-canal-treated teeth without interappointment intracanal medicament are much higher than in those with intracanal medication, indicating maximized bacterial reduction when using antibacterial medicaments before obturation [11]. Calcium hydroxide, the most commonly used intracanal medicament, demonstrates limited antimicrobial activity, while triple-antibiotic paste shows a greater antibacterial effect compared with calcium hydroxide [12,13]. Although there is no scientific evidence supporting the topical application of antibiotics for root canal disinfection, antibiotics are indeed used in some situations, such as regenerative endodontic procedures and tooth replantation after avulsion [12]. The main concern is that the unnecessary use of antibiotics may cause antibiotic resistance. Therefore, a novel antibacterial medicament with a minimal risk of developing microbial resistance may be a viable way to treat *E. faecalis* infections.

Teixobactin, a novel antibiotic, is highly effective against a wide range of Gram-positive bacteria, such as methicillin-resistant *Staphylococcus aureus*, vancomycin-resistant *Enterococcus* spp., and multidrug-resistant *Mycobacterium tuberculosis* [14,15]. It can specifically bind to the bacterial-cell-wall precursors lipid II and lipid III on the cell membrane; therefore, the cell wall integrity is severely disrupted, leading to autolytic cell death [14,16]. However, teixobactin cannot permeate the lipopolysaccharide component of the Gram-negative bacterial cell wall, thereby limiting its use in targeting Gram-negative bacteria [17]. Due to its unique mode of action towards the biosynthetic cell wall building blocks and reduced horizontal gene transfer of the target pathogens, bacterial resistance to teixobactin has not been found in in vitro studies [14,16,18]. Since its discovery, investigations have continued toward the synthesis and modification of the teixobactin structure [17]. *_L_*-Chg_10_-teixobactin, one of the most potent and promising teixobactin analogues, was recently synthesized for combating Gram-positive bacteria [18]. To date, there have been no investigations undertaken to determine the antibacterial effect of *_L_*-Chg_10_-teixobactin against *E. faecalis*, either in the planktonic or biofilm state. In this study, we aimed to investigate the antibacterial and antibiofilm efficacy of *_L_*-Chg_10_-teixobactin against *E. faecalis* in the planktonic and biofilm states.

## 2. Materials and Methods

### 2.1. Preparation of Inoculum

Two different strains of *E. faecalis*, ATCC 29212 (isolated from the urinary tract) and ATCC 47077 (isolated from the deep dentine caries), were obtained from the Central Research Laboratory of the Dental Faculty. Both strains were anaerobically cultivated the frozen stock cultures in brain heart infusion (BHI) broth (Oxoid Ltd., Basingstoke, Hants, UK) at 37 °C for 24 h (Forma Anaerobic System, Thermo Fisher Scientific Ltd., South San Francisco, CA, USA). Each broth culture was then diluted with BHI broth until a turbidity of 0.5 on the McFarland scale, verified with a DU 730 UV/Vis spectrophotometer (Beckman Coulter, Inc. Brea, CA, USA), was obtained.

### 2.2. Preparation of Drugs

A stock suspension of 1 mg/mL *_L_*-Chg_10_-teixobactin was prepared by dissolving 1 mg of *_L_*-Chg_10_-teixobactin powder in 1 mL of 5% dimethyl sulfoxide (DMSO) (5% *v*/*v*) (Sigma Aldrich; St. Louis, MO, USA). Ampicillin (Sigma Aldrich) was used as the positive control. A stock suspension of 0.8 mg/mL ampicillin was prepared by dissolving 0.8 mg of ampicillin powder in 1 mL of 5% DMSO (5% *v*/*v*), based on the reported MIC values of ampicillin ranging from 0.5 to 8 μg/mL used to treat *E. faecalis* [19]. DMSO solvent and BHI groups were used as the negative control groups to rule out any potential antimicrobial effects exerted by these solutions and to ensure no occurrence of contamination.

### 2.3. Minimum Inhibitory Concentration (MIC)

The MICs of *_L_*-Chg_10_-teixobactin and ampicillin were determined by utilizing microtiter 96-well plates (Costar, Corning Incorporated Life Science, Wujiang, China) and a two-fold serial dilution of stock solution using the BHI broth microdilution technique. Briefly, 100 μL of bacterial suspension (10^6^ CFU/mL) was transferred into each well of the microtiter 96-well plate. Ten microliters of the drug suspension was added to each well, but not to the negative control group, to obtain a total volume of 110 μL. Finally, each well contained either *_L_*-Chg_10_-teixobactin ranging from 0.2 to 100 μg/mL or ampicillin ranging from 0.2 to 80 μg/mL. The 96-well plates were then anaerobically incubated at 37 °C for 24 h in an atmosphere of 10% CO_2_, 10% H_2,_ and 80% N_2_ (Forma Anaerobic System, Thermo Fisher Scientific Ltd.).

Turbidity was measured by the optic absorbance density (OD) generated by bacteria that caused light scattering. Higher turbidity reflected a higher number of *E. faecalis* cells in the individual well, and vice versa. The MIC was determined as the lowest concentration of *_L_*-Chg_10_-teixobactin and ampicillin at which no visible turbidity was observed. In this study, OD values were measured at 660 nm by a microplate reader (SoftMax pro-6.3.1, SpectraMax M2; Molecular Devices, Sunnyvale, CA, USA). The reduction rate of OD was calculated using the biofilm inhibition formula:(1)$$\mathrm{OD}\;\mathrm{inhibition}\;\mathrm{rate}=\frac{\left({\mathrm{OD}}_{\mathrm{untreated}}-{\mathrm{OD}}_{\mathrm{blank}}\right)-\left({\mathrm{OD}}_{\mathrm{treated}}-{\mathrm{OD}}_{\mathrm{blank}}\right)}{\left({\mathrm{OD}}_{\mathrm{untreated}}-{\mathrm{OD}}_{\mathrm{blank}}\right)}\times 100\%$$

### 2.4. Minimum Bactericidal Concentration (MBC)

The MBC was measured as the lowest concentration of *_L_*-Chg_10_-teixobactin and ampicillin capable of achieving a 99.9% reduction in viable bacterial growth on the sterile blood agar plates. One hundred microliters of the bacterial suspension (10^6^ CFU/mL) was subcultured onto the agar plates. Subsequently, 10 μL of *_L_*-Chg_10_-teixobactin and ampicillin of different concentrations, i.e., MIC × 2 and MIC × 4, and a lower concentration, i.e., MIC/2, were added and anaerobically incubated at 37 °C for 24 h as aforementioned.

### 2.5. Minimum Biofilm Inhibitory Concentration (MBIC)

The MBIC was calculated as the lowest concentration that inhibited at least 80% of the biofilm biomass growth. The standard safranin colorimetric assay was used as reported previously [20]. *E. faecalis* ATCC 47077 strain was used. Serial dilutions of *_L_*-Chg_10_-teixobactin were carried out with BHI broth to produce final concentrations of 0.8 (MIC), 0.4 (MIC/2), 0.2 (MIC/4), and 0.1 μg/mL (MIC/8). We transferred 20 μL of the *_L_*-Chg_10_-teixobactin from each concentration into the microtiter 96-well plate, followed by the addition of 200 μL of bacterial inoculum suspension (10^6^ CFU/mL), to obtain a total volume of 220 μL. The plates were then aerobically incubated at 37 °C for 24 h. The supernatants from all wells were discarded at the end of the culture period and washed three times with sterile phosphate-buffered saline (PBS) to remove the planktonic cells. Biofilms were stained with 200 μL of 0.1% safranin solution and were subjected to further incubation for 30 min at room temperature. After incubation, the excess stain was removed by washing three times with PBS, followed by 10 min of air drying. Finally, wells were incubated for 15 min with 33% acetic acid per well to dissolve the stain, and 100 μL was pipetted from each well into a new microtiter plate. The OD was measured at a wavelength of 492 nm in a microplate reader (SoftMax pro-6.3.1, SpectraMax M2, Molecular Devices Corp., Sunnyvale, CA, USA) to quantify the biomass.

### 2.6. Minimum Biofilm Eradication Concentration (MBEC)

**Viability assay:** The analysis of the effect of *_L_*-Chg_10_-teixobactin on the metabolic activity of the 24 h old *E. faecalis* biofilms was carried out with a 2,3-*bis*(2-methoxy-4-nitro-5-sulfo-phenyl)-2*H*-tetrazolium-5-carboxanilide (XTT) metabolic activity test (Sigma Aldrich). Firstly, 100 μL (10^6^ CFU/mL) of *E. faecalis* ATCC 47077 strain in BHI broth was aerobically cultured in a 96-well plate at 37 °C for 24 h. The stock solution of *_L_*-Chg_10_-teixobactin was calculated as 32 MIC (25.6 μg/mL). Serial dilutions of the stock solution were carried out to produce 16 MIC (12.8 μg/mL), 8 MIC (6.4 μg/mL), 4 MIC (3.2 μg/mL), 2 MIC (1.6 μg/mL), and MIC (0.8 μg/mL). Twenty microliters of each concentration was added into the 24 h old biofilms. After 24 h, the wells were rinsed three times with PBS and stained with 200 μL XTT (1 mg/mL of XTT in PBS containing 4 M menadione) for 3 h in the dark at 37 °C. Subsequently, 100 μL of the solution was transferred to a new plate and the OD was measured with a microplate reader at 490 nm. The same formula for the calculation of MBIC was used for obtaining the MBEC. The MBEC_50_, MBEC70, and MBEC80 were calculated as the concentrations of *_L_*-Chg_10_-teixobactin that could reduce the biomass viability by 50%, 70%, and 80%, respectively.

**Biomass assay:** The ability of *_L_*-Chg_10_-teixobactin to treat the 24 h old *E. faecalis* biofilm for 24 h was further measured by the standard safranin colorimetric assay to assess the biofilm biomass formation.

All of the experiments were independently repeated in triplicate for MIC and MBC and in quadruplicate for MBIC and MBEC. Ethics committee approval and informed consent were not required because the experiments did not involve human or animal studies. This study was conducted in accordance with the Declaration of Helsinki.

### 2.7. Statistical Analysis

All data were analyzed using Excel spreadsheets (version 16.51, Microsoft^®^ excel. 2021, Microsoft Corporation, Redmond, WA, USA) and are presented as mean ± standard deviation (SD). Log dose–response curves were plotted to calculate the nonlinear data for optimal MBIC and MBEC.

## 3. Results

### 3.1. Minimum Inhibitory Concentration (MIC)

The MIC values for all the tested groups against *E. faecalis* were calculated using the percentage of the optical density of the microtiter plate (Figure 1 and Table 1). *_L_*-Chg_10_-teixobactin was effective against *E. faecalis* with MIC values of 0.8 μg/mL for both strains. The MIC value of ampicillin ranged from 1.25 to 2.5 μg/mL for ATCC 47077 and was 1.25 μg/mL for ATCC 29212; these values were at least 1.5-fold greater than that of *_L_*-Chg_10_-teixobactin. DMSO solvent showed no antimicrobial effect on both strains of *E. faecalis*.

### 3.2. Minimum Bactericidal Concentration (MBC)

The MBC values of *_L_*-Chg_10_-teixobactin against ATCC 29212 and ATCC 47077 strains were found to be 0.8 μg/mL, which are equivalent to the MIC values (Figure 1 and Table 1). The MBC values of ampicillin against ATCC 29212 and ATCC 47077 were 10 and 20 μg/mL, respectively, which are 12-fold greater than that of *_L_*-Chg_10_-teixobactin.

### 3.3. Minimum Biofilm Inhibitory Concentration (MBIC)

When a subminimal inhibitory concentration of *_L_*-Chg_10_-teixobactin was added to the *E. faecalis* ATCC 47077 strain, the safranin–biomass assay showed that 0.1 μg/mL *_L_*-Chg_10_-teixobactin inhibited 78.16% of the biomass formation (Figure 2). Nonlinear data analysis using the log dose-response curve demonstrated that two concentrations of *_L_*-Chg_10_-teixobactin, 0.13 and 0.23 μg/mL, were able to reduce 80% of the biomass. Hence, the lowest concentration of 0.13 μg/mL was defined as the optimum MBIC80.

### 3.4. Minimum Biofilm Eradication Concentration (MBEC)

The viability assay shows a wide range of concentrations that can decrease the viability of the preformed biofilm (Figure 3A). Nonlinear data analysis using the log dose -response curve showed that 1.1 μg/mL *_L_*-Chg_10_-teixobactin reduced 50% of the biomass formation (MBEC_50_) (Figure 3B). The MBEC70 and MBEC80 of *_L_*-Chg_10_-teixobactin were 1.6 and 4.6 μg/mL, respectively. Different concentrations of *_L_*-Chg_10_-teixobactin resulted in a 36.55% to 68.33% biomass reduction (Figure 3C). Nonlinear data analysis using a log dose-response curve showed that the minimum concentration required to achieve a 50% reduction in the biomass formation was 1.1 μg/mL (Figure 3D).

## 4. Discussion

*_L_*-Chg_10_-teixobactin is a novel teixobactin analogue that claims to exert a potent antibacterial effect by targeting nonendogenous proteins through binding to the conserved motif of lipid II and lipid III of the bacterial cell wall, disrupting the biosynthesis of peptidoglycan [16,17,18,21]. To date, the antibacterial efficacy of *_L_*-Chg_10_-teixobactin against *E. faecalis* has not been explored. The latter is characterized by its ability to cause biofilm-induced infectious diseases and increase resistance to multiple antibiotics, and by its implication in endodontic treatment failure [1,4,19]. Investigations of the antibacterial effect of *_L_*-Chg_10_-teixobactin on *E. faecalis* may provide a glimpse of its potential application to serve as an antibacterial agent in endodontic treatment. However, endodontic diseases are polymicrobial infections [3,4], indicating the need to further study other common endodontic pathogens and multispecies biofilms. Furthermore, teixobactin is only effective at killing Gram-positive bacteria, leaving Gram-negative bacteria unscathed [14,15,17]. Thus, consideration should be given to combining teixobactin and its analogues with another antimicrobial compound that is effective against Gram-negative bacteria to prevent selection for their survival.

In this study, two different strains of *E. faecalis,* ATCC 29212 and ATCC 47077, were selected because they have been extensively used as representative strains in clinical and laboratory trials [19,22]. To eliminate any possibility of cross-contamination by cross-talk between the bacterial cells of different strains, each strain was cultured in different microtiter 96-well plates. The reliability of the results was enhanced by calculating the average values obtained from three independent repetitions of the experiments in triplicate for MIC and MBC, and in quadruplicate for MBIC and MBEC. DMSO does not demonstrate any antimicrobial effect; thus, it was used as the solvent to prepare both the *_L_*-Chg_10_-teixobactin and ampicillin.

MIC is defined as the lowest concentration required for the antibacterial agents to inhibit ≥90% of bacterial growth [23,24]. In this study, the MIC value of *_L_*-Chg_10_-teixobactin on *E. faecalis* ATCC 29212 and ATCC 47077 strains was lower than the 2 μg/mL reported by Darnell et al. [25] on a different strain of *E. faecalis*, i.e., strain JH2-2; it was slightly higher than the 0.5 μg/mL reported by Ling et al. [16] on vancomycin-resistant *E. faecalis* [16,25]. Two plausible explanations may account for the differences, including the use of different teixobactin or its analogues as the tested drug and the different strains of *E. faecalis* studied. Parmer et al. [26,27] reported a broad range of MIC values ranging from 0.25 to 128 μg/mL when different types of teixobactin analogues were tested against *S. aureus*. Our results showed that the MIC values of ampicillin were at least 1.5-fold greater than those of *_L_*-Chg10-teixobactin. However, the MIC value of ampicillin appears to be strain-dependent, as observed in our study, with higher concentrations required to inhibit the growth of ATCC 47077 compared with ATCC 29212. This concurred with the findings of another study reporting that the MIC values for ampicillin were higher (i.e., ranging from 4 to 8 μg/mL) when it was tested against the *E. faecalis strain* that was ampicillin-resistant, compared with the MIC values of 0.5 to 2 μg/mL when it was tested against the nonresistant strain counterpart [19]. The MBC is the lowest concentration of antibiotics required to kill >99.9% of bacteria [28]. This study found that the MBC values of *_L_*-Chg10-teixobactin were much lower than those obtained from ampicillin. Tong et al. [22] also reported similarly increased MBC values of ampicillin on ATCC 29212 and ATCC 47077 strains as 16 and 32 mg/mL, respectively [22].

Bacteria within biofilms can be 10- to 1000-fold less susceptible to various antimicrobial agents than their planktonic counterparts; so, it is logical to study the antimicrobial effects of *_L_*-Chg_10_-teixobactin on *E. faecalis* biofilm biomass [29,30]. In the present study, the *E.*
*faecalis* ATCC 47077 strain was selected because it was isolated from deep dentine caries and should resemble the strains commonly found in infected root canals. It is a plasmid-free strain with an intrinsic and acquired resistance to antibiotics, a strong ability to survive, and the virulence features that can be found in most *E. faecalis* genotypes [31,32]. We observed that *_L_*-Chg_10_-teixobactin inhibited 80% of biofilm formation at a concentration of 0.13 μg/mL; this is considerably lower than the MBICs of ampicillin (8.19 μg/mL) and vancomycin (4.10 μg/mL), as reported by Sandoe et al. [33]. Hence, we extrapolated that *_L_*-Chg_10_-teixobactin should exhibit greater potency toward the inhibition of *E. faecalis* biofilms. A further experiment was conducted in this study to investigate the eradicating effect of *_L_*-Chg_10_-teixobactin on the 24 h old biofilms using an XTT assay. The concentration of *_L_*-Chg_10_-teixobactin required to eradicate 80% of the 24 h old *E. faecalis* biofilms was relatively lower than the reported MBECs of many other antibiotics [33]. The results suggested that *_L_*-Chg_10_-teixobactin demonstrates the potential to outperform other antimicrobial agents in eradicating *E. faecalis* biofilms. The MBEC80 of *_L_*-Chg_10_-teixobactin was approximately 6 MIC, which echoes the finding of Hussein et al., which showed that 4 MIC of the teixobactin analogue was effective in suppressing the growth of *S. aureus* [34].

To date, although bacterial resistance to teixobactin and its analogues has not been found in in vitro studies, possibly attributed to their nonprotein drug target [14,16,18], antibiotic resistance cannot be completely ruled out; many bacterial species that may carry intrinsic resistance have not yet been tested, while others may develop acquired resistance through mechanisms of horizontal gene transfer or mutation to their DNA chromosomes. Despite the limitations of this in vitro study, *_L_*-Chg_10_-teixobactin exhibited stronger antibacterial and antibiofilm effects against the two strains of *E. faecalis* studied at very low concentrations when compared with ampicillin. Further research is warranted to investigate its efficacy against matured *E. faecalis* biofilms, other common endodontic pathogens, and multispecies biofilms.

## 5. Conclusions

In conclusion, *_L_*-Chg_10_-teixobactin displayed potent antimicrobial activity against the prevalent endodontic pathogen found in failed endodontic cases, i.e., *E. faecalis* in both planktonic and biofilm states, at very low concentrations. Therefore, *_L_*-Chg_10_-teixobactin may be used as an effective antibacterial agent in endodontic treatment.

## Figures and Tables

**Figure 1 microorganisms-10-01099-f001:**
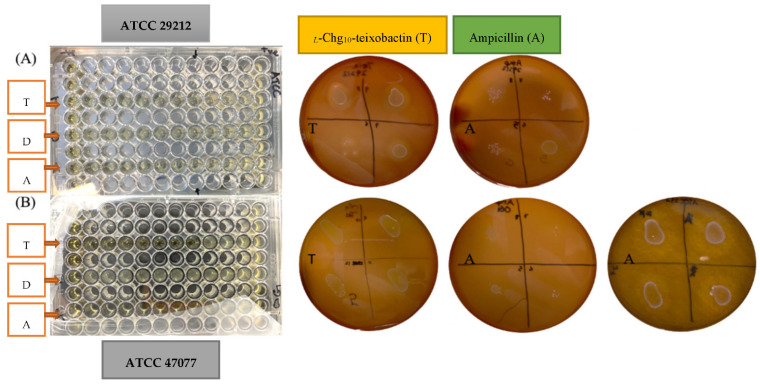
The microtiter 96-well plate (**left**) and the agar plates (**right**) used in the experiments for the measurement of MIC and MBC against the two strains of *E. faecalis*. *_L_*-Chg_10_-teixobactin (T), DMSO (D), and ampicillin (A) were labelled on both the 96-well plate and the agar plates that were tested against: top row (**A**) ATCC 29212 strain; bottom row (**B**) ATCC 47077 strain.

**Figure 2 microorganisms-10-01099-f002:**
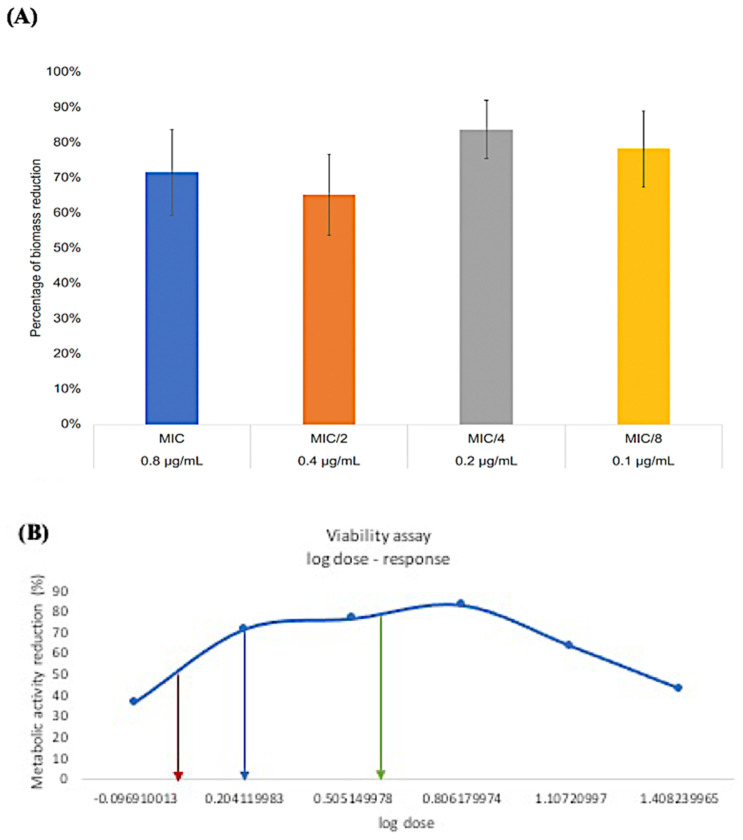
(**A**) The inhibition rate of *E. faecalis* biofilm (MBIC) under *_L_*-Chg_10_-teixobactin treatments of different concentrations, i.e., MIC (0.8 μg/mL), MIC/2 (0.4 μg/mL), MIC/4 (0.2 μg/mL), and MIC/8 (0.1 μg/mL); (**B**) the log dose-response curve demonstrating two MBICs, i.e., 0.13 and 0.23 μg/mL, that achieved 80% reduction of *E. faecalis* biofilm (green arrow).

**Figure 3 microorganisms-10-01099-f003:**
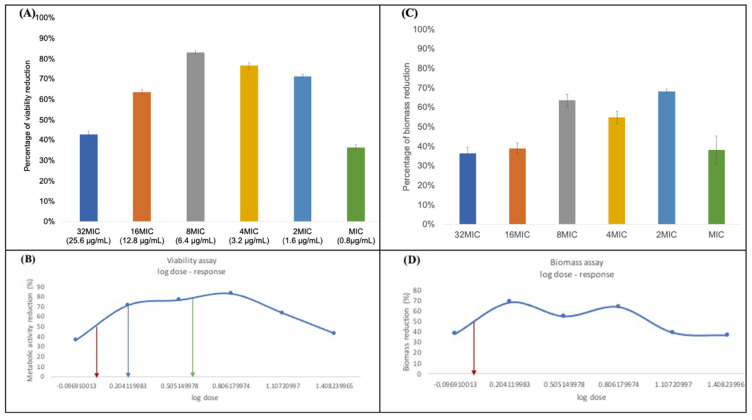
(**A**) The results of the viability assay of 24 h old *E. faecalis* biofilm (MBEC) under *_L_*-Chg_10_-teixobactin treatments with different concentrations, i.e., 32 MIC (25.6 μg/mL), 16 MIC (12.8 μg/mL), 8 MIC (6.4 μg/mL), 4 MIC (3.2 μg/mL), 2 MIC (1.6 μg/mL), and MIC (0.8 μg/mL); (**B**) the log dose -response curve demonstrates the MBEC_50_ (1.1 μg/mL), MBEC_70_ (1.6 μg/mL), and MBEC_80_ (4.6 μg/mL) required to achieve a 50% (red arrow), 70% (blue arrow), and 80% (green arrow) reduction in the metabolic activity of *E. faecalis*, respectively; (**C**) the results of the safranin assay used to measure the reduction in 24 h old *E. faecalis* biofilm under *_L_*-Chg_10_-teixobactin treatments with different concentrations as stated in (**A**); (**D**) the log dose-response curve demonstrates that the minimum concentration required for *_L_*-Chg_10_-teixobactin to achieve 50% reduction in the formation of *E. faecalis* biofilm was 1.1 μg/mL (red arrow).

**Table 1 microorganisms-10-01099-t001:** MIC and MBC for *_L_*-Chg_10_-teixobactin and ampicillin (μg/mL).

	Teixobactin Analogue		Ampicillin		DMSO	
Treatment Group	MIC	MBC	MIC	MBC	MIC	MBC
ATCC 29212	0.8	0.8	1.25	10	N/A	N/A
ATCC 47077	0.8	0.8	1.25-5	20	N/A	N/A

## Data Availability

Data are available from the first author A.J.

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
