# Peer review of "Antimicrobial Effects of L-Chg10-Teixobactin against Enterococcus faecalis In Vitro"

_microorganisms, 2022, doi:10.3390/microorganisms10061099_

Round 1
Reviewer 1 Report
Thanks to choose MDPI and microorganism to publish you manuscript.
line 20 Citation: ABSTRACT ?? need authors.
line 28 keyword in bold.
The structure of the papaer needs some precautions, in the initial part there are totally missing authors and affiliations.
the quality of Figure 2 is not optimal
Miss author contribution
Author Contributions: For research articles with several authors, a short paragraph specifying their individual contributions must be provided. The following statements should be used “Conceptualization, X.X. and Y.Y.; methodology, X.X.; software, X.X.; validation, X.X., Y.Y. and Z.Z.; formal analysis, X.X.; investigation, X.X.; resources, X.X.; data curation, X.X.; writing—original draft preparation, X.X.; writing—review and editing, X.X.; visualization, X.X.; supervision, X.X.; project administration, X.X.; funding acquisition, Y.Y. All authors have read and agreed to the published version of the manuscript.” Please turn to the CRediT taxonomy for the term explanation. Authorship must be limited to those who have contributed substantially to the work reported.
Some typo also in references
newspaper in italics and year in bold
- Author 1, A.B.; Author 2, C.D. Title of the article. Abbreviated Journal Name Year, Volume, page range.
From the scientific point of view, materials and methods and discussions, I find the manuscript very interesting and valid for the journal.
Many other studies can be conducted in this field for the sake of our patients to avoid antibiotic resistance; topic much discussed in the last decade
Author Response
Dear Reviewer,
Thank you very much for your time in reviewing our manuscript and the constructive comments given. Please find the point-to-point response to your comments as follows:
Point 1: line 20 Citation: ABSTRACT ?? need authors.
Response 1: ‘Authors’ and ‘title’ of article have been added to Line 20 Citation.
Point 2: line 28 keyword in bold.
Response 2: Line 28 ‘Keyword’ is now in bold.
Point 3: the quality of Figure 2 is not optimal
Response 3: Figure 2 has been replaced with a higher quality image.
Point 4: The structure of the papaer needs some precautions, in the initial part there are totally missing authors and affiliations.
Response 4: Statement of “Author Contributions” has been added to P9 line 373-378 according to the template given by you, i.e., "AUTHOR CONTRIBUTIONS: conceptualization, Zhang C and Li X; methodology, Zhang C, Prasanna N and Jarkhi A; software, Hu M; validation, Jarkhi A and Hu M; formal analysis, Jarkhi A; investigation, Jarkhi A; resources, Sun Z and Li X; data curation, Jarkhi A; writing—original draft preparation, Jarkhi A and Lee A; writing—review and editing, Lee A, Sun Z and ZhangC; visualization, Zhang C; supervision, Zhang C, Prasanna N and Lee A; project administration, Jarkhi A; funding acquisition, Zhang C. All authors have read and agreed to the published version of the manuscript."
Point 5: Some typo also in references - newspaper in italics and year in bold: Author 1, A.B.; Author 2, C.D. Title of the article. Abbreviated Journal Name Year, Volume, page range.
Response 5: All the references have been revised so that journal is in italics and year is in bold.
I sincerely look forward to any further comments you may have in regards to the revised manuscript.
Once again, thank you very much!
Best regards,
Prof C.F. Zhang (on behalf of all authors)
Reviewer 2 Report
The present manuscript depicts some antimicrobial results of L-Chg10-teixobactin against two Enterococcus faecalis strains. The topic is interesting. However, there are some major points which have to be addressed:
1) Endodontic infections are usually caused by more than one species. It would have been sufficient to test more bacterial species. This has not been discussed sufficiently by the authors.
2) The authors should not repeat their results in the dicussion. Especially the conclusions should be rewritten without depicting any exact values of the results.
3) Teixobactin kills only Gram-positive bacteria. Hence, its use as single coumpound for endodontic treatment is not sufficient. A combination with other antimicrobial compounds which are active against Gram-negative bacteria is meaningful. This point should be discussed.
4) What about antibiotic resistance? can it be really excluded as stated by the authors? This point deserves also more discussion.
5) The legend of Figure 1 should be depicted in more detail
Minor points
1) Bacterial names should be written in italic throughout the manuscript (see page 2, lines 63-64)
2) Page 2, line 64: Enterococcus spp.
Author Response
Dear Reviewer 2,
Thank you for your time in reviewing this manuscript and the constructive comments provided. Please see the point-to-point response to your comments as follows:
Point 1: Endodontic infections are usually caused by more than one species. It would have been sufficient to test more bacterial species. This has not been discussed sufficiently by the authors.
Response 1: This point has been added to the ‘discussion’ in P7 line 262-264, “However, endodontic diseases are polymicrobial infection (3, 4), indicating the need to elaborate research on other common endodontic pathogens and multi-species biofilm.”
Point 2: The authors should not repeat their results in the dicusson. Especially the conclusions should be rewritten without depicting any exact values of the results.
Response 2:
- Discussion has been revised to ensure no repetition of the results.
- Conclusion has been revised without depicting the exact values of the results in P9 line 368-370.
Point 3: Teixobactin kills only Gram-positive bacteria. Hence, its use as single coumpound for endodontic treatment is not sufficient. A combination with other antimicrobial compounds which are active against Gram-negative bacteria is meaningful. This point should be discussed.
Response 3: This point has been added to the ‘discussion’ in P7 line 264-268 “…..Furthermore, teixobactin is only effective in killing gram-positive bacteria, leaving gram-negative bacteria unscathed (14, 15, 17). Thus, consideration should be given to combining teixobactin and its analogues with another antimicrobial compound that is effective against gram-negative bacteria to prevent selection for their survival.”
Point 4: What about antibiotic resistance? can it be really excluded as stated by the authors? This point deserves also more discussion.
Response 4: This point has been added in P8 line 320-325…..”Although bacterial resistance to teixobactin and its analogues have not been found in the in vitro studies possibly attributed to their non-protein drug target (14, 16, 18), antibiotic resistance cannot be completely ruled out; as many bacterial species that may carry intrinsic resistance have not yet been tested, while others may develop acquired resistance through mechanisms of horizontal gene transfer or mutation to its own DNA chromosomes. “
Point 5: The legend of Figure 1 should be depicted in more detail
Response 5: The legend of Figure 1 has been depicted in more detail to describe the experiments done, i.e. MIC on micrometer 96-well plate and MBC on agar plate; and the two strains of E. faecalis tested, i.e. top row - ATCC 29212 strain, and bottom row - ATCC 47077 strain. Labels had also been added to Figure 1 to improve clarity.
Minor points:
Point 6: Bacterial names should be written in italic throughout the manuscript (see page 2, lines 63-64)
Response 6: Bacteria names in page 2, lines 63-64 have been changed to italic, i.e., “…..methicillin-resistant Staphylococcus aureus, vancomycin-resistant Enterococcus, and multidrug-resistant Mycobacterium tuberculosis.”
Point 7: Page 2, line 64: Enterococcus spp.
Response 7: Page 2, line 64: vancomycin-resistant Enterococcus has been changed to Enterococcus spp.
Once again, I really appreciate your comments and feedback, and shall look forward to any further comments you may have in regards to the revised manuscript.
Thanks in advance.
Best regards
Prof C.F. Zhang (on behalf of all authors)
Round 2
Reviewer 2 Report
All points have been revised sufficiently
Author Response
Dear Reviewer
Once again, thank you very much for your invaluable comments.
The statement from P8 Line 292:
"This concurred with the findings of another study that showed higher MIC values obtained for the ampicillin on ampicillin-resistant E. faecalis strain ranging from 4 to 8 μg/mL, compared to the 0.5 to 2 μg/mL obtained for the non-resistant strain (19)."
....has been rephrased as follows (highlighted as yellow):
"This concurred with the findings of another study reporting that the MIC values for ampicillin were higher (i.e., ranging from 4 to 8 μg/mL) when it was tested against the E. faecalis strain that was ampicillin-resistant, compared to the MIC values of 0.5 to 2 μg/mL when tested against the non-resistant strain counterpart (19)."
The entire manuscript has also been carefully proofread to ensure no spelling and grammatical mistakes.
Best regards
Prof C.F. Zhang
Corresponding author (on behalf of all authors)